# The Current Understanding of Molecular Mechanisms in Adenomyosis-Associated Infertility and the Treatment Strategy for Assisted Reproductive Technology

**DOI:** 10.3390/ijms25168937

**Published:** 2024-08-16

**Authors:** Hsien-Ming Wu, Tian-Chi Tsai, Shang-Min Liu, Angel Hsin-Yu Pai, Liang-Hsuan Chen

**Affiliations:** 1Department of Obstetrics and Gynecology, Linkou Medical Center, Chang Gung Memorial Hospital, Taoyuan 333, Taiwan; danielwu@cgmh.org.tw (H.-M.W.); mp2186@cgmh.org.tw (T.-C.T.); h8e0n0r7y18@gmail.com (S.-M.L.); iaplegna@gmail.com (A.H.-Y.P.); 2Graduate Institute of Clinical Medical Sciences, College of Medicine, Chang Gung University, Taoyuan 333, Taiwan

**Keywords:** adenomyosis, infertility, endometrial receptivity, embryo implantation, pregnancy, assisted reproductive technology

## Abstract

Adenomyosis, endometriosis of the uterus, is associated with an increased likelihood of abnormal endometrial molecular expressions thought to impair implantation and early embryo development, resulting in disrupted fertility, including the local effects of sex steroid and pituitary hormones, immune responses, inflammatory factors, and neuroangiogenic mediators. In the recent literature, all of the proposed pathogenetic mechanisms of adenomyosis reduce endometrial receptivity and alter the adhesion molecule expression necessary for embryo implantation. The evidence so far has shown that adenomyosis causes lower pregnancy and live birth rates, higher miscarriage rates, as well as adverse obstetric and neonatal outcomes. Both pharmaceutical and surgical treatments for adenomyosis seem to have a positive impact on reproductive outcomes, leading to improved pregnancy and live birth rates. In addition, adenomyosis has negative impacts on reproductive outcomes in patients undergoing assisted reproductive technology. This association appears less significant after patients follow a long gonadotropin-releasing hormone agonist (GnRHa) protocol, which improves implantation rates. The pre-treatment of GnRHa can also be beneficial before engaging in natural conception attempts. This review aims to discover adenomyosis-associated infertility and to provide patient-specific treatment options.

## 1. Introduction

Adenomyosis is an estrogen-dependent uterine disorder characterized by ectopic endometrial-like tissue (stroma, glands, and fibroblasts) pathologically demonstrated in the myometrium, causing hyperplasia and hypertrophy in the surrounding smooth muscle cells and local inflammatory responses [1]. Women affected by adenomyosis may present with an enlarged uterus, heavy menstrual bleeding, dysmenorrhea, dyspareunia, chronic pelvic pain, and infertility, but one-third of them are asymptomatic [2].

Historically, adenomyosis was diagnosed by a histopathological finding after hysterectomy [3]. The epidemiological scenario has changed; introducing new medical compounds and surgical techniques has allowed clinicians to treat the disease conservatively. A 20.9% prevalence of adenomyosis in the general population was shown from ultrasound units [4], whereas the data range from 10% to 35% in histological reports after hysterectomy [5]. Noninvasive methods, such as ultrasounds and magnetic resonance imaging (MRI), now allow for the clinical diagnosis of adenomyosis in infertile women [6,7], leading to a better understanding of the epidemiology, natural history, and consequences on fertility and obstetric outcomes. Four subtypes of adenomyosis assessed by MRI were classified as follows: subtypes I-III were suggested as causes of direct endometrial invasion, ectopic endometriotic invasion from the outside, and de novo metaplasia, respectively, whereas subtype IV was a heterogeneous mixture of far advanced disease [8]. Although the most common risk factor profile [9] included an age of more than 40 years, multiparity, prior cesarean delivery, or uterine surgery [9,10], over the past decade, adenomyosis has been increasingly identified in young fertile-aged women [11], in infertility patients [12], and those with pain, abnormal uterine bleeding, or both [13]. Because of recent advancements in imaging techniques [14], improved diagnostic tools have raised awareness of the condition in adenomyosis-associated infertility. Additionally, adenomyosis usually coexists with other gynecological disorders, such as uterine fibroids and endometriosis. Adenomyosis and endometriosis share several features; for a long time, adenomyosis has been called endometriosis interna [15], which the International Statistical Classification of Diseases and Related Health Problems (ICD) coded as “endometriosis of uterus” https://www.icd10data.com/ICD10CM/Codes/N00-N99/N80-N98/N80-/N80.0 (accessed on 1 October 2023). Although they often coexist in the same patients, they are considered two distinct entities due to different pathogenic pathways and clinical characteristics [16,17].

Among women receiving assisted reproductive technology (ART), the prevalence of endometriosis is widely variable, ranging from 20% to 80% [16,17], whereas in those with a history of adenomyosis, it is 20% to 25% [12]. Endometriosis influences up to 40% of infertile women and is associated with poorer ART outcomes, including a decreased yield of mature oocytes [18], lower implantation rates [19,20], and reduced pregnancy rates [19,20]. Adenomyotic foci impair the molecular factors of embryo attachment and endometrial decidualization, resulting in altered expressions of endometrial receptivity genes, growth factors, cytokines, and myometrial contractility in a fashion that disrupts implantation [21,22]. Recently, the development of imaging resolution has led to an increased number of women being diagnosed with adenomyosis, strengthening the interest in more conservative treatment options [23]. Regarding the scarcity of high-quality studies in fertility-preserving management, a shared guideline of ART strategies will be important in the future as the disease requires a lifelong plan for symptom control and reproductive outcomes. This review aims to discover adenomyosis-associated infertility and to provide clarified guidance for managing infertile women with adenomyosis.

## 2. Pathophysiologic Aspects of Adenomyosis-Associated Infertility

Adenomyosis is the presence of ectopic endometrial glands and stroma surrounded by hypertrophic and hyperplastic myometrium, subsequently establishing dysfunction in embryo transport and implantation [21,22]. The pathogenesis of adenomyosis remains unclear; in the past decade, several hypotheses have been proposed [24], and an increasing number of studies have demonstrated that sex steroid hormone receptors, inflammatory molecules, extracellular matrix enzymes, growth factors, and neuroangiogenic factors act as pathogenic mediators of adenomyosis [21].

### 2.1. Myometrium Structure and Physiology in Fecundity

Histologically, the myometrium comprises an outer longitudinal layer and an inner circular layer of smooth muscle cells between the endometrium and uterine serosa [25]. Differing from other mucosal tissues (e.g., intestine), the endometrial–myometrial interface (EMI) is a mucosal–muscular interface without an intervening basement membrane, where the endometrial basalis directly contacts the myometrium [26,27] (Figure 1). MRI provides definitions of high-signal intensity endometrium, medium-signal intensity outer myometrium, and low-signal intensity inner myometrium (also called the “junctional zone”) on T2-weighted images [28]. With a poor histological correlation, the junctional zone is identified in MRI studies of the uterus as the subendometrial halo or in ultrasonography as the hypoechoic tissue identified beyond the endometrial basal layer [28,29]. Additionally, their embryologic origins and physiological roles differ markedly. During embryogenic development, the junctional zone arises from the Mullerian ducts, as does the endometrium, whereas the outer myometrium is mesenchymal in origin. The inner myometrium, endowed with estrogen and progesterone receptors, displays cycle-dependent directional contractions throughout menstruation [30,31]. In the follicular phase, retrograde contractions (uterine cervix to fundus) of the inner myometrium facilitate sperm transport. The contractile frequency and amplitude decrease markedly in the mid-luteal phase to promote embryo implantation. During menses, antegrade-propagated (uterine fundus to cervix) contractions with increased peristalsis amplitude assist the desquamation of the shed endometrium [30,32]. In contrast, the outer myometrium protects the fetus throughout pregnancy and mechanically facilitates the expulsion of the fetus at parturition under the regulation of oxytocin and steroid hormones [33,34].

### 2.2. Theories and Potential Mechanisms of Adenomyosis

#### 2.2.1. Invagination Theory

As one of the most accepted theories, adenomyotic lesions grow from an enhanced invasion of the endometrium into the myometrium through an interrupted or absent junctional zone [35] (Figure 2). Studies have postulated the dysregulation of extracellular matrix function in the eutopic endometria of women with adenomyosis compared with healthy controls, facilitating endometrial cell proliferation, epithelial-to-mesenchymal transition (EMT), resistance to apoptosis, migration and invasion into the myometrium, and the establishment of adenomyosis [36,37]. Excessive estradiol in the eutopic endometria of women with adenomyosis drives molecular mechanisms of proliferation and apoptosis dysfunctionalities in adenomyotic lesions [37,38,39]. EMT, a process wherein epithelial cells acquire an invasive and metastatic phenotype, is characterized by the loss of E-cadherin and enhanced mesenchymal marker expression, which occur concomitantly with the up-regulation of N-cadherin and vimentin in the eutopic and ectopic endometrium of patients with adenomyosis compared with healthy controls [40], contributing to endometrial cell migration into the myometrium. Platelet aggregation [39] and high estrogen levels [40] have been considered possible causes of endometrial EMT in the development of adenomyosis. Moreover, the dysregulation of extracellular cell matrix function, which is mediated by the downregulated Lysyl oxidase gene [41] or the up-regulated matrix metalloproteinases [42,43], presented in the eutopic endometrium in women with adenomyosis, supporting the invagination theory.

#### 2.2.2. Microtrauma of Endometrial–Myometrial Interface

The theory of tissue injury and repair (TIAR) as the primary mechanism in the initiation process of adenomyosis [44] stresses the importance of tissue damage in the endometrial–myometrial interface (EMI). In support of this hypothesis are the clinical observations that adenomyosis is associated with multiparity, repeated endometrial curettage, previous cesarean section, and prior uterine surgery wherein the endometrial–myometrial interface is breached [24,45]. However, nulliparous women without a history of uterine surgeries also develop adenomyosis. The proposal is that physiologic trauma (“microtrauma”) to the EMI results from the course of proliferation and inflammation at the EMI by chronic uterine auto-traumatization (tissue injury) and, subsequently, tissue repair [14]. Continuous cyclic uterine peristaltic activity induces repeated cycles of auto-traumatization, damaging the EMI and migrating basal endometrium fragments into the myometrium throughout a woman’s reproductive lifetime [46]. Increased estrogen levels in the eutopic endometrium promote microtrauma to the EMI following TIAR and the peristaltic activity of the subendometrial myometrium [47]. Studies have demonstrated that microtrauma activates the TIAR process and produces prostaglandin E2 (PGE2) through interleukin-1 (IL-1)-induced cyclo-oxygenase-2 (COX-2) expression, facilitating local estrogen production through STAR and aromatase P450 [22]. While increased estrogen promotes healing through estrogen receptor β, it also supports oxytocin-mediated hyperperistalsis through estrogen receptor α, which inhibits the healing course and augments further microtrauma to EMI, eventually leading to the establishment of adenomyotic lesions [14,47].

#### 2.2.3. De Novo Metaplasia Theory

The alternative hypothesis of adenomyosis proposes a de novo process of embryonic or adult endometrial stem cell metaplasia into the myometrium [48]. During Müllerian duct development and fusion, some embryonic pluripotent Müllerian remnants may be misplaced in the myometrium, subsequently resulting in metaplastic changes in the adult myometrium and the development of de novo adenomyotic foci [49]. Furthermore, endometrial epithelial progenitor and mesenchymal stem cells have been reported in the endometrial basalis, playing a critical role in the cyclic repair of the endometrium [50]. These adult stem cells can be activated after tissue injury at the EMI, promoting uncontrolled growth into the myometrium and differentiating into adenomyotic lesions [21].

#### 2.2.4. Outside-to-Inside Invasion

Apart from invasion directly into the myometrium, adult endometrial and stromal stem cells may be deposited into the myometrium through retrograde menstruation [24]. Chapron et al. proposed the “outside-to-inside invasion” theory that endometrial cells in retrograde menstrual blood potentially infiltrate pelvic organs and the uterine serosa [15]. The posterior focal adenomyosis of the outer myometrium in patients with deep infiltrating endometriosis nodules in the posterior compartment supports this hypothesis. Similarly, through outside-to-inside trans-serosa invasion, the high concurrence of anterior focal adenomyosis of the outer myometrium and deep infiltrating endometriosis nodules in the vesicouterine pouch have been reported [51].

Nevertheless, existing theories may not fully explain the heterogeneous phenotypes of adenomyosis. Most importantly, elucidating the adenomyosis pathogenesis may help us better understand clinical symptoms and the association of imaging presentations.

### 2.3. Pathogenesis of Adenomyosis

#### 2.3.1. Genetic and Epigenetic Alteration

Thousands of abnormally up- or downregulated genes have been reported in the eutopic endometria of women with adenomyosis compared to controls [41]. Studies have found that the variants of the Cytochrome P450 (CYP) gene and catechol-O-methyltransferase (COMT) gene affect enzyme activity and promote estrogen-dependent diseases, including adenomyosis [52,53]. Current evidence has demonstrated the role of genetic polymorphisms involved in the adenomyosis pathogenesis through alterations in the metabolism and functions of steroid hormones and their receptors [54,55], extracellular matrix dysregulation [56], angiogenesis [57], and inflammatory mediators [58]. A recent next-generation sequencing study detected KRAS and PIK3CA mutations in adenomyotic lesions and the adjacent eutopic endometrium, suggesting that these mutations most likely arose in the eutopic endometrial epithelial cells before they invaded the myometrium, promoting invasive and survival capacities that facilitate their invasion and growth within the myometrial tissue, thereby establishing adenomyosis [59].

Epigenetic abnormalities can be equally applied to adenomyosis, including DNA methylation, histone modification, and microRNA expression, which give rise to aberrant mRNA and the protein expression of genes [21]. The aberrant expression of deoxyribonucleic acid methyltransferases (DNMTs), a family of enzymes that catalyze the transfer of a methyl group to DNA, was detected in patients with adenomyosis compared with controls [60]. The hypomethylation of DNA and increased expression of the CCAAT/enhancer-binding protein β, a transcription factor regulating cellular proliferation and differentiation, have been demonstrated in adenomyosis [61]. The promoter hypermethylation of progesterone receptor β can lead to progesterone resistance in adenomyotic tissue [62]. The increased expression of class I histone deacetylases (HDACs) was also associated with the development of adenomyosis [63]. Methyltransferase-like 3 (METTL3), a writer promoting the methylation of RNA, and the total N6-methyladenosine (m6A) levels were decreased in the eutopic endometria of patients with adenomyosis compared with controls, suggesting that m6A RNA methylation regulators may be involved in the pathogenesis of adenomyosis [64].

#### 2.3.2. Hyperestrogenism and Progesterone Resistance

As an estrogen-dependent uterine disorder, adenomyosis is associated with local steroid hormone aberration. Aromatase enzyme activity and protein expression are up-regulated in adenomyotic tissue [65]. The epigenetic-regulated NR5A1 (a nuclear receptor) binding to the proximal promoter of the aromatase (CYP19A1) gene seems to be the critical mechanism for its suppression in normal endometrial stromal cells and its expression in endometriotic stromal cells [66]. Moreover, a lower expression of 17-β hydroxysteroid dehydrogenase type 2 (HSD17-β2), which converts estradiol (E2) to the less potent estrone (E1), was detected in the eutopic and ectopic endometria of patients with adenomyosis compared with controls [67]. Lastly, increased biosynthesis and decreased conversion of E2 contribute to a local hyperestrogenic milieu in adenomyosis, promoting the excessive production of prostaglandins driven by estrogen receptor β that causes an inflammatory process [52].

The expression of uterine estrogen and progesterone receptors in patients with adenomyosis differs from that in healthy controls [68]. Estrogen receptor β (ERβ) is overexpressed in adenomyotic tissue during the proliferative phase and throughout the myometrium across the menstrual cycle. In contrast, compared with controls, estrogen receptor α (ERα) is downregulated in the eutopic endometria of patients with adenomyosis during the secretory phase. Additionally, lower progesterone receptor (PRα and PRβ) expression is discovered in adenomyotic tissue as a clue of progesterone resistance [69,70].

#### 2.3.3. Immune Disorder and Inflammatory Mediators

Specific human leukocyte antigen (HLA) classes and increased numbers of macrophages and other immune cells in the endometria of women with adenomyosis have been proposed to activate the autoimmune system, causing a dysregulated immune response [71,72]. Altered COX-2 and PGE2 synthesis, correlating with an increased expression of corticotropin-releasing hormone (CRH), facilitates the inflammatory pathway [73]. The increased expression of inflammatory mediators, including IL-1β, the IL-18/IL-18R complex, and tumor necrosis factor-α,β (TNF-α,β), in the endometria of women with adenomyosis further implicate the activation of the nuclear factor kappa light-chain enhancer of activated B cell (NF-κB) pathways in developing adenomyotic lesions [74,75].

#### 2.3.4. Neuroangiogenesis and Fibrosis

Particularly in symptomatic women, higher expressions of neurogenic factors, including nerve growth factors (NGFs), synaptophysin (SYN), and microtubule-associated protein 2 (MAP2), have been reported in adenomyotic lesions compared with controls [76]. Inflammatory mediators, known as PGE2 and prostacyclin, may stimulate sensory unmyelinated C nerve fibers in the functional layer of the endometrium, causing neurogenic inflammation in adenomyotic lesions. The glycoprotein neural cell adhesion molecule, such as CD56, and other neurotrophic factors found in the endometrium and adenomyotic lesions further stimulate nerve growth, increasing the sensitivity of nociceptors in the myometrium, thus triggering pelvic pain [77,78].

The overexpression of angiogenic factors, such as VEGF, annexinA2 (ANXA2), follistatin, and activin A, members of the TGF-β family, has been detected in the ectopic and eutopic endometria of patients with adenomyosis, promoting vascular sprouting and augmenting vascular permeability in angiogenesis [79,80,81]. Due to repeated cycles of TIAR, TGF-β signaling drives EMT and collagen production, ultimately leading to a certain degree of fibrosis in adenomyosis [39].

#### 2.3.5. Pituitary Hormones and Endocrine-Disrupting Chemicals

Prolactin (PRL) expression has increased in women with adenomyosis compared to those without adenomyosis [82]. In animal studies, hyperprolactinemia-triggered adenomyosis was demonstrated in a mouse model [83,84], and the overexpression of PRL receptor mRNA was detected in the uteri of mice with versus those without adenomyosis [84]. PRL directly influences the myometrium and induces adenomyosis by the degeneration of smooth muscle cells [85]; however, the mechanisms still await further delineation.

The oxytocin receptor (OTR) is expressed in the normal endometrium and myometrium and is regulated by the cyclic steroid hormone [86]. The overexpression of OTR in adenomyosis-affected uteri may induce hyperperistalsis and microtrauma in the EMI, thus positively correlating with the severity of the disease. Moreover, there is a significantly higher expression of myometrial OTR in the fundal region of adenomyosis-affected uteri than in the isthmic region, causing disorientation of the EMI contractions and subsequently interfering with sperm transport and fertility [87].

Endocrine-disrupting chemicals (EDCs), such as estrogen-mimetic bisphenol A and diethylstilbestrol, have been reported as the effects of gene–environment interaction during embryogenesis and neonatal development in animal studies [88,89], giving rise to adenomyosis development. Furthermore, phthalates, a kind of EDC that have antiandrogenic effects, are reported to increase the risk of adenomyosis [90].

#### 2.3.6. Microbiota Composition

An alteration in the microbiota composition has been postulated in the pathogenesis of adenomyosis by affecting the host’s epigenetic, immunologic, metabolic, and biochemical functions. Distinct bacterial taxa were identified in the endometrial, vaginal, and gut microbiota of patients with adenomyosis compared with the controls, suggesting a potential link between the microbiota profile and adenomyosis [91].

## 3. Pathologic Mechanisms in Adenomyosis-Associated Infertility

Adenomyosis has been regarded as a typical uterine disorder in multiparous women; however, increasing evidence declares an association with infertility and reproductive failure [92]. Recently, a cross-sectional study showed that the prevalence of adenomyosis was 24.4% in infertile women over 40 years old and 22% in those under 40 years old. The percentage increased to 38.2% in women with recurrent miscarriages and 34.7% with previous artificial reproductive treatment failure [12]. The effects of adenomyosis on eutopic endometrial function, uterine environment, and contractility alter the capacity to receive the embryo and accept proper development, causing infertility and poor implantation outcomes [12,93] (Figure 3). However, it is not easy to precisely understand the specific role of adenomyosis due to the frequent coexistence of endometriosis and adenomyosis [94].

### 3.1. Oocyte Quality and Embryo Euploidy

An impaired follicular environment has been advocated in endometriosis, which makes oogenesis more susceptible to meiotic errors and chromosomal instability [19]. Toxic pelvic factors directly and persistently affect the quality of the gametes in the local environment and temporarily affect the embryos while passing through the fallopian tube. Some studies demonstrated that an alteration in the meiotic spindle could increase the embryo aneuploidy rate and thus cause higher miscarriage rates and pregnancy losses in patients with endometriosis [95,96]. In ART, in bypassed tubal and pelvic environments, Juneau et al. [97] reported equal euploidy rates in women with endometriosis and age-matched controls. Regardless of the high correlation with endometriosis, the negative impact on oocytes in women with adenomyosis remains elusive.

### 3.2. Utero-Tubal Transport

As described earlier, the opposite pattern of OTR expression in fundal and isthmic regions could disturb the direction of EMI contractions, interfering with gametes or embryo transport and subsequent implantation into the endometrium [87]. Additionally, adenomyosis seems to destroy the normal architecture of the myometrium, resulting in the anatomical distortion of the uterine cavity and disturbing uterine peristalsis [98].

### 3.3. Endometrial Receptivity

The implantation process requires coordinated and synchronous embryo development and a decidualized endometrium receptive to implantation [99]. Aberrant endometrial shifts during the window of implantation (WOI), including impaired decidual transformation, increased inflammatory mediators, the dysregulation of immune factors, and oxidative stress, ultimately contributed to defective endometrial receptivity. Dysregulated implantation-associated factors, such as HOXA10 [100,101], the leukemia inhibitory factor [102], NR4A and FOXO1A [103], and MMP2 [42], leading to a shift in gene expression during the WOI, have been reported to promote progesterone resistance and impair decidualization. The decreased expression of cell adhesion receptors, including glycodelin, the integrin family, Mucin-1, and osteopontin, further reduces embryo-to-endometrial interaction in implantation [104,105].

Steroid metabolism and biosynthesis in the endometrium also alter endometrial receptivity. The overexpression of aromatase, coupled with the downregulation of the E2 metabolizing enzyme HSD17-β2, enhances bioavailable E2 in the endometrium, accounting for aberrantly high estrogen receptor levels and proliferation, interfering embryo attachment and invasion [65]. E2-regulated COX-2 and hypoxia-induced factor-1 are involved in inflammation associated with implantation failure [106].

Recently, Khan et al. [107] demonstrated microvilli malformation and axonemal arrangement disruption in the apical endometria derived from women with focal and diffuse adenomyosis. These ultra-structural endometrial abnormalities in response to tissue inflammation with abundant macrophage infiltration may negatively affect fertility outcomes in women with symptomatic adenomyosis.

In addition, women with adenomyosis have a higher concurrent rate of chronic endometritis [108]. A variable presence of chronic endometritis in women with adenomyosis could be an additional factor that worsens the endometrial receptivity by distorting the functional and structural integrity of the endometrium [109].

### 3.4. Immunotolerance and Angiogenesis

During the WOI, natural killer (NK) cells and macrophages are the dominant immune cells that regulate trophoblast invasion macrophages [110]. In addition, the vascular endothelial growth factor and placental growth factor are also important in promoting angiogenesis [111] in early pregnancy. In endometriosis, activated macrophages can secrete pro-inflammatory cytokines, such as IL-1β, IL-6, IL-8, interferon-γ (IFN-γ), and TNF-α in the peritoneal cavity [112] and pro-inflammatory cytokines that trigger venous thromboembolism during pregnancy [113]. Several mechanisms appear to be implicated in the association between adenomyosis and obstetric complications, including activated local and systemic inflammation, increased prostaglandin production in myometrium, aberrant uterine contractility, and defective spiral artery remodeling at the basis of placentation, resulting in several detrimental obstetric consequences, including early pregnancy losses and possibly repeated pregnancy loss (RPL) [114,115].

### 3.5. Uterine Peristaltic Wave

Studies have shown that higher uterine contraction frequencies in natural [116] and stimulated cycles [117] are associated with reduced conception, implantation, and live birth rates. In adenomyosis, ultrastructural myometrial abnormalities distort the uterine cavity and cause a disturbance in the normal rhythmic muscle contractility [118]. In addition, subendometrial myometrium (inner myometrium or junctional zone) presents with dysfunctional hyperperistalsis and increased intrauterine pressure in women with adenomyosis, amplified by local hyperestrogenism, leading to defective implantation and placentation and, in turn, miscarriages and RPL [119].

## 4. Assisted Reproductive Technology for Adenomyosis-Associated Infertility

The management of adenomyosis-associated infertility has been conflicting. The current medical treatments for symptomatic adenomyosis, including nonsteroidal anti-inflammatory drugs (NSAIDs), progestins, oral contraceptives, a levonorgestrel-releasing intrauterine system (LNG-IUS), as well as both gonadotropin-releasing hormone (GnRH) agonists and antagonists, are based on anti-proliferative and anti-inflammatory effects by controlling the hormonal medium. Regardless of the modality, the treatment of adenomyosis can temporarily improve symptoms and quality of life, consistently improving conception chances [120,121,122]. The adverse effects of adenomyosis on ART outcomes have also been reported [92,123,124]. However, there are limited data evaluating the impact of ART with medical or surgical intervention on women with adenomyosis-associated infertility.

### 4.1. Ovarian Stimulation Protocol

According to the pathogenic mechanisms of adenomyosis, several ovarian stimulation protocols have been attempted to improve reproductive outcomes. The use of long-acting or continuous gonadotropin-releasing hormone agonist (GnRHa) administration in an ovarian stimulation protocol, which suppresses premature LH surge and avoids spontaneous ovulation, was more extensive, possibly because of a hypoestrogenic state and an anti-proliferative effect in adenomyotic tissue [125].

Regarding the duration of GnRHa downregulation, the GnRHa protocol was classified as an ultra-long (a monthly long-acting GnRHa injection with a duration varying from one to six cycles), long (a daily dose of GnRHa starting from the previous luteal phase until hCG is triggered), or short (a daily dose of GnRHa given on days 2–4 of the menstrual cycle until hCG is triggered) protocol. In a previous systematic review of 10 studies [126], a pooled analysis showed that a long protocol had better reproductive outcomes than a short protocol regarding the pregnancy, live birth, and miscarriage rates. Several studies have demonstrated positive results with an increased clinical pregnancy rate (CPR) [127] and implantation rate (IR) [127] and a decreased miscarriage rate (MR) [128] in the ultra-long protocol when compared with the long protocol. However, the ultra-long protocol causes the profound inhibition of ovarian function by the downregulation of its receptors, thus reducing the sensitivity of pituitary glands, usually causing an increase in the duration and dosage of gonadotropin, resulting in poor ovarian response and decreased oocyte retrieval [127].

Ge et al. systematically evaluated the reproductive outcomes of different ovarian stimulation protocols in fresh embryo transfer (ET) cycles [129]. The intragroup comparison showed no statistical differences in the number of retrieved oocytes and maturation rate. The results elucidate that an ultra-long or long protocol might benefit women with adenomyosis receiving ART with fresh ET compared to a short protocol. In fresh ET in women aged ≥35, the CPR was higher in the ultra-long and long protocols than in the antagonist and short protocols (52.1%, 50.0% vs. 20.0%, 27.5%; *p* = 0.031) [129]. Nevertheless, embryos derived from protocols, such as long, short, and antagonist protocols, had no impact on pregnancy outcomes in frozen embryo transfer (FET) cycles (Table 1).

### 4.2. Embryo Transfer Strategy

Frozen embryo transfer (FET) is commonly used when the endometrial characteristics are unsuitable for fresh cycle transfer, when there is a high risk of ovarian hyperstimulation syndrome, and when there is further promotion of the cumulative live birth rate [138]. FET with artificial endometrial preparation is widely accepted due to the mechanism that ovarian suppression induced by the hormone replacement regimen possibly plays a role in reducing endometrial alterations linked to ovarian function in adenomyosis [129,130]. With the pre-treatment of long-term GnRHa downregulation, FET was a protective factor for the LBR and a cost-effective method with a lower total gonadotrophin dosage and shorter stimulation duration than fresh ET with an ultra-long protocol [130].

#### 4.2.1. Segmentation of Embryo Transfer

The pre-treatment of hormonal suppression, such as through GnRHa or progestins, 3–6 months before FET, also known as deferred ET or segmental ET, was thought to produce a hypo-estrogen status, causing the regression of adenomyotic lesions and the correction of endometrial alterations, thus improving the ART outcomes. Based on these theories, a freeze-all cycle with a deferred embryo transfer strategy is recommended for patients with endometriosis [139], a treatment of choice that does not increase the risk of endometriosis flare [140,141,142,143]. Studies have shown that reproductive outcomes significantly improved in women with adenomyosis undergoing deferred ET with long-term GnRHa preparation for 2 to 4 months [92,122,134]. Interestingly, Zhang et al. found that the uterine volume did not change before or after the GnRHa treatment (82.0 ± 13.4 cm^3^ vs. 79.3 ± 14.0 cm^3^; *p* = 0.123), suggesting that the downregulation effect of GnRHa may improve pregnancy outcomes in other ways [134]. Also, women undergoing in vitro fertilization (IVF) with LNG-IUS pre-treatment for three months before FET have higher clinical pregnancy, implantation, and ongoing pregnancy rates [131]. Another case report showed that a patient receiving deferred ET after dienogest pre-treatment for three months achieved pregnancy with reduced adenomyotic lesions and serum marker cancer antigen-125 (CA-125) [135].

#### 4.2.2. GnRHa Pre-Treatment

FET following long-term GnRHa pre-treatment has a better ART outcome and a potential benefit in terms of a lower gonadotrophin dose and a shorter stimulation duration than fresh ET combined with a long or ultra-long GnRHa protocol [130]. GnRHa has been found to markedly suppress the hypothalamus–pituitary–ovarian axis and reduce the angiogenesis and inflammatory response of adenomyotic foci, leading to the regression of adenomyosis as well as symptom relief [121,144]. Regarding the effect of GnRHa on endometrial receptivity in women with adenomyosis, Tian et al. proposed a transcriptome analysis of the eutopic endometrium, which identified the systemic role and underlying cellular regulatory mechanisms of GnRHa treatment in adenomyosis-associated infertility [145]. After GnRHa treatment, the study elucidated 132 differentially expressed genes in the eutopic endometria of women with adenomyosis. Chemokine (C-C motif) ligand 21 (CCL21), related to immune system-associated signal transduction, was highly expressed after the GnRHa treatment, indicating that a molecular regulation may be involved in the improvement of endometrial receptivity in adenomyosis. Nevertheless, the routine use of GnRHa before fertility management in infertile women with adenomyosis remains conflicting. According to some retrospective studies [121,129,133,146], pre-treatment with GnRHa before FET has not shown a beneficial effect on IVF outcomes in women with adenomyosis. Moreover, a higher miscarriage rate (MR) was found in women with adenomyosis undergoing IVF with GnRHa pre-treatment [132,136,137]. A recent retrospective study revealed that the GnRHa downregulation group had a larger mean diameter of the uterus and a higher proportion of severe dysmenorrhea at baseline than the non-downregulation group [137]. The pregnancy outcomes had no difference between the two groups, except for a higher MR in the GnRHa downregulation group, suggesting that the MR was difficult to reverse through GnRHa treatment in severe adenomyosis. Randomized controlled trials with more cases should be designed to verify the correlation between a higher MR and GnRHa treatment in adenomyosis. Conflicting results may be attributed to a mixed variety of adenomyosis features without a proper classification and a lack of consensus on ART protocols.

#### 4.2.3. Oxytocin Antagonists

Uterine hyperperistalsis with higher serum oxytocin levels was found in women with endometriosis compared to healthy controls [147,148], which may contribute to the development of endometriosis as well as to adverse effects on IVF outcomes in women with endometriosis. The increased immunoreactivity of oxytocin receptors has been found in eutopic endometrial stromal and epithelial cells and in myometrial and vascular cells in ectopic adenomyosis foci, correlating with the severity of dysmenorrhea [149]. Atosiban, an oxytocin receptor antagonist, can decrease myometrial contractility and preferentially relax uterine arteries, enhancing endometrial perfusion and supporting embryo implantation [150]. A randomized controlled study has shown that Atosiban treatment before FET provides better clinical efficacy in the priming of the uterus in patients with endometriosis [147]. Lin et al. reported that atosiban increased the pregnancy rate among patients with endometriosis without coexisting adenomyosis but not for those with coexisting adenomyosis [151]. They stratified the analyses by adenomyosis features and found an insignificant benefit of atosiban therapy among patients with more extensive adenomyosis. Further evidence is required to prove that atosiban has a therapeutic role in adenomyosis-associated infertility.

### 4.3. Uterus Sparing Surgery

Fertility-preserving adenomyomectomy plans to balance the advantages of removing the pathology against the disadvantages of possibly destroying the uterine structure. Proper conservative surgery seems to restore fertility in women with adenomyosis, as successful pregnancies have been reported in a systematic review [126]. Many methods and techniques have been demonstrated for proper excision of the affected area and reconstruction of the myometrial defect, either by laparoscopy, laparotomy, or hysteroscopy.

In a systematic review study, Grimbizis et al. reported pregnancy rates of 60.5% after complete excision, 46.9% after partial excision of adenomyosis, and 55.6% after non-excisional techniques [152]. Rocha et al. reported a higher spontaneous pregnancy rate in the group with GnRHa for 24 weeks after surgery compared with surgery alone (40.7% vs. 15.0%; *p* = 0.002) [126]. Younes et al. reported that the odds ratio of clinical pregnancy rates after surgery for adenomyosis was 6.22 (95% CI 2.34–16.54) in a meta-analysis [92] and reported successful pregnancies after fertility-preserving surgery in three-fourths of the cases treated for adenomyosis in another review [153]. Tan et al. compared the reproductive outcomes in diffuse versus focal adenomyosis following surgical or combined medical management. Their results show that women with focal adenomyosis have higher pregnancy rates (54.8% versus 31.3%), live birth rates (45.1% versus 23.8%), and surprisingly higher miscarriage rates (21.8% versus 19.6%) than those with diffuse adenomyosis [154]. In a recent meta-analysis, Jiang suggested excisional treatment as a consideration for patients with symptomatic adenomyosis and infertility for several years or repeated failure with ART [155]. However, there is a lack of consensus on the rationale for removing the adenomyotic foci to improve fertility due to the high heterogenicity between studies. The surgeon’s experience is important in managing an optimal surgical method for adenomyosis; several surgical techniques (open or laparoscopic), excision techniques (complete or partial adenomyomectomy), and modified closure techniques (U-shaped suturing, overlapping flaps, double-/triple-flap method, and transverse H-incisions) have been proposed [156,157].

Fertility-preserving surgical treatment for adenomyosis is associated with a higher risk of adverse obstetric outcomes in future pregnancies. The removal of adenomyotic foci with significant amounts of myometrium may cause the formation of uterine scars and a reduction in uterine capacity, which results in the uterus having a lack of ability to grow during pregnancy, thus increasing the risk of uterine rupture. The risk of uterine rupture during pregnancy is more than 1.0% versus 0.26% in women receiving uterine adenomyomectomy and myomectomy, respectively [158]. Otsubo et al. suggested that preserving a 9 to 15 mm uterine wall thickness after an adenomyotic foci excision is safe for future pregnancies [159]. In addition, the destruction of the endometrium, together with the junctional zone, may cause serial obstetric complications, including miscarriage, preterm birth, and placentation disorders. Although the presented studies favored the safety of the obstetric outcomes in women treated with adenomyomectomy, it is uncertain whether surgery is involved in better reproductive performance or not.

## 5. Prognostic Tools for Adenomyosis-Associated Infertility

An increasing number of studies have reported the impact of adenomyosis features on successful pregnancy outcomes after fertility treatment, including the extent or severity of adenomyosis lesions, based on images and clinical presentation. As adenomyosis becomes more prevalent in older women, a degree of confounding factors, such as declining ovarian reserve and an older age, harms reproductive outcomes.

### 5.1. Imaging Features of Adenomyosis

Adenomyosis is expressed as several morphologic characteristics, including asymmetrical myometrial thickening, intramyometrial cysts or hyperechoic islands, linear striae, fan-shaped shadowing, myometrial echogenic subendometrial lines and buds, translesional vascularity, and the disruption of the endometrial myometrial junction, which increases the magnitude of the adverse effect [160]. With the evolution of imaging techniques, such as high-resolution ultrasound and magnetic resonance imaging (MRI), we can now diagnose adenomyosis relatively reliably in women undergoing conservative treatments. Although no uniform standardized reporting system exists for image-based adenomyosis diagnostic criteria, the International Morphological Uterus Sonographic Assessment (MUSA) group agreed on terminology for describing myometrial lesions in ultrasonography. The sonographic features of adenomyosis could be described and classified according to their lesion size, location, differentiation (focal/diffuse), appearance (cystic/non-cystic), uterine myometrium involvement (depth of invasion), disease extent (<25%, 25–50%, and >50% of uterine volume), and number of foci [161]. Recently, several studies proposed a positive correlation between the cumulative impact of ultrasound parameters and the clinical features of adenomyosis [162].

Furthermore, the presence of numerous sonographic features of adenomyosis worsens fertility outcomes. According to a multicenter prospective study, the clinical pregnancy rate decreased from 42.7% in the control group to 22.9% and 13.0% in women with four and seven ultrasound diagnostic features of adenomyosis, respectively [163]. Additionally, the focal type of adenomyosis seems to have a more significant negative effect than other types [128,164]. A significant lower cumulative pregnancy rate and severe obstetric complications, such as placenta previa, placenta accreta, preeclampsia, and preterm birth, were associated with adenomyosis located in the posterior uterine wall, possibly due to a higher concurrence rate of ovarian endometrioma, pelvic adhesion, and revised American Society for Reproductive Medicine (rASRM) scores [165]. However, a recent systematic review and meta-analysis revealed that women with focal adenomyosis do not significantly affect the reproductive outcome compared to those with diffuse lesions [124]. A validated model that includes sonographic predictors and symptoms of adenomyosis may enrich clinical practice regarding adenomyosis.

### 5.2. Serum Marker

The serum marker cancer antigen 125 (CA-125) is positively correlated with the severity of adenomyosis. It indicates concomitant mixed endometriosis, suggesting that CA-125 may be a valuable indicator for monitoring the efficacy of adenomyosis treatment and future fertility outcomes [166,167,168]. A retrospective study of women with severe adenomyosis treated with a six-month course of GnRH agonist following conservative surgery showed that the severity of dysmenorrhea was significantly improved with a decline in the serum level of CA-125 [167]. A higher spontaneous pregnancy rate was found in women with a postoperative serum level of CA-125 of less than 10.00 IU/mL. Another retrospective study found that patients with a greater than sevenfold decrease in the CA-125 level after GnRH agonist treatment might have better IVF outcomes [166]. Chang et al. showed that patients with adenomyosis who had a successful live birth during the three-year follow-up after surgical–medical therapy tended to be younger and nulliparous with a lower body mass index, lower baseline analgesic usage score, lower preoperative serum CA-125, and with an anterior adenoma location [168]. However, to evaluate the pregnancy-associated factors, a stepwise multivariate linear regression analysis showed that only age and the baseline analgesic usage score were independent predictors of future pregnancy in women with adenomyosis who underwent treatment.

## 6. Conclusions

Adenomyosis, defined as endometrial cell invasion into the myometrium, usually presents with pelvic pain, heavy menstrual bleeding, and reproductive failure, which may be associated with abnormal local effects of sex steroid and pituitary hormones, immune response, inflammatory factors, and neuroangiogenic mediators. The proposed pathogenetic mechanisms affect the uterine cavity’s receptivity and the endometrial molecular expression necessary for embryo implantation, resulting in decreased pregnancy rates, higher miscarriage rates, and obstetric complications. According to current evidence, the management of adenomyosis, either through medical or surgical approaches, may positively affect reproductive outcomes. Women receiving assisted reproductive treatments with a downregulation protocol or segmental embryo transfer appear to have better pregnancy results. More research is necessary to evaluate the management of adenomyosis-associated infertility and provide treatment guidance for the challenging condition of adenomyosis.

## Figures and Tables

**Figure 1 ijms-25-08937-f001:**
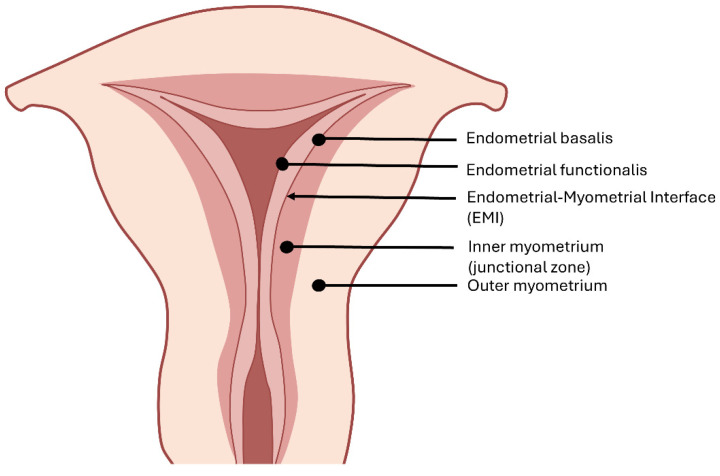
The structure of the uterus. The endometrial–myometrial interface (EMI) is a mucosal–muscular interface where the endometrial basalis directly contacts the myometrium.

**Figure 2 ijms-25-08937-f002:**
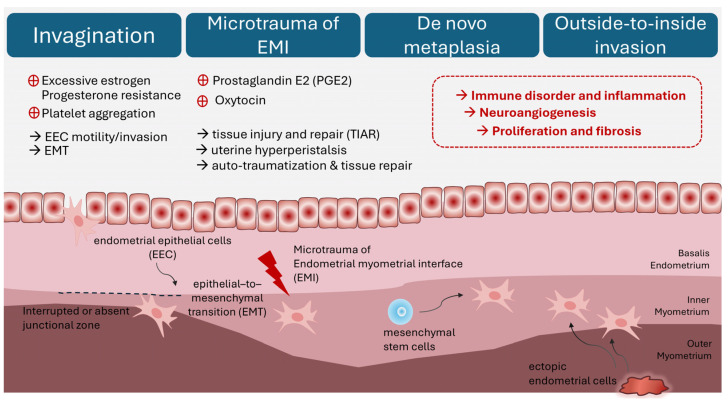
Theories and potential mechanisms of adenomyosis. (1) Invasion of endometrial basalis into myometrium. (2) Microtrauma of endometrial–myometrial interface. (3) De novo metaplasia from stem cells. (4) Outside-to-inside invasion induced by ectopic endometrial cells.

**Figure 3 ijms-25-08937-f003:**
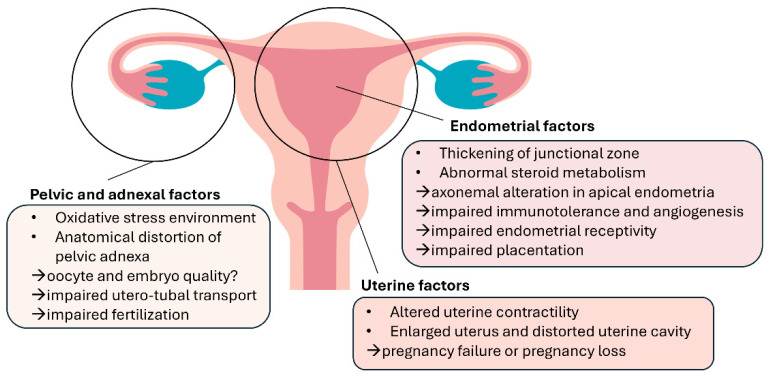
Pathologic mechanisms in adenomyosis-associated infertility.

**Table 1 ijms-25-08937-t001:** Ovarian stimulation protocol and embryo transfer strategy for adenomyosis.

Ovarian Stimulation Protocol		
Study	Intervention	Results
Rocha et al. (2018) [126]	Analysis of 10 adenomyosis studies: Pooled short protocols (*n* = 785) Pooled long protocols (N = 482)	A long protocol had better outcomes than a short protocol in CPR (43.3% vs. 31.8%; *p* = 0.0001), LBR (43% vs. 23.1%; *p* = 0.005), and MR (18.5% vs. 31.1%; *p* < 0.0001).
Hou et al. (2020) [127]	Observational cohort study: Controls in long protocol (N = 3471) Patients in long protocol (N = 127) Patients in ultra-long protocol (N = 362)	Patients with adenomyosis had significantly lower CPRs, IRs, and LBRs in the long protocol than the controls. In patients with adenomyosis, the CPR (OR 1.925, 95% CI 1.137–3.250; *p* = 0.015), IR (OR 1.694, 95% CI 1.006–2.854; *p* = 0.047), and LBR (OR 1.704, 95% CI 1.012–2.859; *p* = 0.044) were significantly increased in the ultra-long protocol than in the long protocol.
Lan et al. (2021) [128]	Retrospective study: Ultra-long GnRHa protocol (N = 212) Long GnRHa protocol (N = 116)	There was a significantly lower MR in the ultra-long GnRHa group than in the long protocol (12.0% vs. 26.5%; *p* = 0.045). There were no differences in the IR, CPR, or LBR.
Wu et al. (2022) [130]	Observational cohort study: Group A: FET with GnRHa pre-treatment (N = 192) Group B: fresh ET with ultra-long protocol (N = 241) Group C: fresh ET with long protocol (N = 104)	The IR and LBR were higher in Group A than in Groups B and C, with significantly lower total gonadotrophin dose and stimulation duration. The IR (OR 1.729, 95% CI 1.073–2.788; *p* = 0.025), CPR (OR 1.665, 95% CI 1.032–2.686; *p* = 0.037), and LBR (OR 1.694, 95% CI 1.045–2.746; *p* = 0.033) increased, and the MR (OR 0.203, 95% CI 0.078–0.530; *p* = 0.001) decreased in Group A compared to Group C. In comparing Groups A and B, FET was a protective factor for the LBR (OR 1.350, 95% CI 1.017–1.792; *p* = 0.038).
Ge et al. (2023) [129]	Retrospective study: Total of 257 fresh cycles with ultra-long (N = 108), long (N = 56), short (N = 59), and antagonist (N = 34) protocols Total of 305 FET cycles with embryos from ultra-long (N = 98), long (N = 101), short (N = 52), and antagonist (N = 54) protocols	In fresh ET cycles, compared with ultra-long and long protocols, the IR (49.7%, 52.1% vs. 28.2%; *p* = 0.001) and CPR (64.3%, 57.4% vs. 35.6%; *p* = 0.004) significantly decreased in the short protocol. In the FET cycles, there were no statistical differences in the IR, CPR, or LBR of embryos derived from different stimulation protocols.
Embryo Transfer Strategy		
Study	Intervention	Results
Niu et al. (2013) [122]	Retrospective study: FET with (N = 194) and without (N = 145) GnRHa pre-treatment	There were significantly higher CPRs (51.35% vs. 24.83%), IRs (32.56% vs. 16.07%), and OPRs (48.91% vs. 21.38%) in the GnRHa pre-treatment group.
Park et al. (2016) [121]	Retrospective study: Group A: fresh ET without GnRHa pre-treatment (N = 147) Group B: fresh ET with GnRHa pre-treatment (N = 105) Group C: FET with GnRHa pre-treatment (N = 43)	The CPR in Group C (39.5%) tended to be higher than those in Group B (30.5%) and Group A (25.2%), but there was no significant difference.
Liang et al. (2019) [131]	Retrospective study: Total of 134 received LNG-IUS (Mirena; Bayer) before FET; 224 controls	There was significantly higher OPRs (41.8% vs. 29.5%; *p* = 0.017), IRs (32.1% vs. 22.1%; *p* = 0.005), and CPRs (44% versus 33.5%; *p =* 0.045) in the LNG-IUS group.
Chen et al. (2020) [132]	Retrospective study: Long GnRHa protocol with (N = 48) and without (N = 140) GnRHa pre-treatment	In fresh ET, the non-pre-treatment group had a higher LBR (37.7% vs. 21.2%, *p =* 0.028) and CLBR (40.50% vs. 27.90%, *p =* 0.019) than the GnRHa pre-treatment group.
Li et al. (2021) [133]	Retrospective study: FET with (N = 160) and without (N = 181) GnRHa pre-treatment (73.8% for 1 month, 15.6% for 2 months, and 10.6% for ≥3 months)	No differences were found in the CPRs (40.63% vs. 42.54%; *p* = 0.72), LBRs (23.75% vs. 23.75%; *p* = 0.74), miscarriage rates, ectopic pregnancy rates, and preterm birth rates.
Zhang et al. (2022) [134]	Retrospective study: FET with (N = 45) and without (N = 218) GnRHa pre-treatment	There were significantly lower MRs (12.5% vs. 37.2%; *p* = 0.044) and higher LBRs (46.7% vs. 24.8%; *p* = 0.009) in the GnRHa pre-treatment group.
Feng et al. (2022) [135]	Case report: FET with GnRHa pre-treatment for 2 months and = second FET with dienogest pre-treatment for 3 months	The CA-125 level and adenomyoma size were markedly reduced after the GnRHa and dienogest pre-treatments. A singleton pregnancy was achieved at the second FET (with the dienogest pre-treatment).
Yang et al. (2023) [136]	Retrospective study: 2048 FET divided into 4 groups: GnRHa-HRT, HRT, OI, and NC (Subgroups in GnRHa-HRT protocol: one, two, and three or more GnRHa injections)	No statistical differences in pregnancy outcomes were found among the four endometrial preparation protocols. In the GnRHa-HRT protocol, the early MR was increased in the subgroup of two compared with one GnRHa injection (18% vs. 6.5%; *p* = 0.017). No differences were found in the CPR or LBR among the subgroups.
Ge et al. (2023) [129]	Retrospective study: Total of 305 FET cycles with or without GnRHa pre-treatment	For women ≥35 years, the IR and CPR were increased in the GnRHa pre-treatment group without statistical differences.
Li et al. (2023) [137]	Retrospective study: Matched 272 cycles in non-downregulation group and 272 in downregulation group (Subgroups in downregulation group: GnRHa for 1 month, 2 months, or ≥3 months)	The pregnancy outcomes in the downregulation group were similar to those in the non-downregulation group, but there was a higher MR (13.4% vs. 3.1%; *p =* 0.003). The subgroups in fresh ET indicated that the IR (75.0% vs. 39.2%; *p =* 0.002) and CPR (83.3% vs. 47.0%; *p =* 0.016) could be improved by prolonged GnRHa downregulation (≥3 months), whereas a late MR was difficult to reverse (30.0% vs. 3.2%; *p =* 0.017).

CPR: clinical pregnancy rate; LBR: live birth rate; MR: miscarriage rate; IR: implantation rate; OR: odd ratio; CI: confidence interval; ET: embryo transfer; FET: frozen embryo transfer; HRT: hormonal replacement therapy; OI: ovulation induction; NC: natural cycle.

## Data Availability

Not applicable.

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
