# Peer review of "The Current Understanding of Molecular Mechanisms in Adenomyosis-Associated Infertility and the Treatment Strategy for Assisted Reproductive Technology"

_ijms, 2024, doi:10.3390/ijms25168937_

Round 1

Reviewer 1 Report

Comments and Suggestions for Authors

The manuscript is overall well-written and organized.

The reviewer suggests some additional discussions that may improve the text..

Comments

1. Introduction

Classification by Kishi et al. can be mentioned. Kishi Y, Suginami H, Kuramori R, Yabuta M, Suginami R, Taniguchi F. Four subtypes of adenomyosis assessed by magnetic resonance imaging and their specification. Am J Obstet Gynecol 2012;207:114.e1–7.

2.2. Theories and potential mechanisms of adenomyosis

The recent unique “Axonemal Alteration Hypothesis” (Khan KN, Fujishita A, Suematsu T, Ogawa K, Koshiba A, Mori T, et al. An axonemal alteration in apical endometria of human adenomyosis. Hum Reprod 2021;36:1574–89) may be introduced.

2.2. Theories and potential mechanisms of adenomyosis

Recent studies suggest that Fusobacterium infection in endometrial stromal fibroblasts is a potential cause of endometriosis, an adenomyosis-related disease (Muraoka A, Suzuki M, Hamaguchi T, Watanabe S, Iijima K, Murofushi Y, Shinjo K, Osuka S, Hariyama Y, Ito M, Ohno K, Kiyono T, Kyo S, Iwase A, Kikkawa F, Kajiyama H, Kondo Y. Fusobacterium infection facilitates the development of endometriosis through the phenotypic transition of endometrial fibroblasts. Sci Transl Med. 2023 Jun 14;15(700):eadd1531

Microbiome studies that compared the gut, uterine cavity, and vaginal cavity between adenomyosis and non-adenomyosis patients can be quoted and discussed.

2.2. Theories and potential mechanisms of adenomyosis

The relationship between adenomyosis and chronic endometritis (Khan KN, Fujishita A, Ogawa K, Koshiba A, Mori T, Itoh K, et al. Occurrence of chronic endometritis in different types of human adenomyosis. Reprod Med Biol 2022;21:e12421. ; Kitaya K, Matsubayashi H, Yamaguchi K, Nishiyama R, Takaya Y, Ishikawa T, et al. Chronic endometritis:Potential cause of infertility and obstetric and neonatal complications. Am J Reprod Immunol 2016;75:13–22.) may be referred.

4.3. Uterus sparing surgery

The article on adenomyomectomy by Nishida M et al., (Masato Nishida, Katsumi Takano, Yuko Arai, Hirokazu Ozone, Ryota Ichika  Fertil Steril. 2010 Jul;94(2):715-9. Conservative surgical management for diffuse uterine adenomyosis) may be worth introducing here as it discloses the surgical methodology with detailed figures.

Author Response

Comment 1: 1. Introduction Classification by Kishi et al. can be mentioned. Kishi Y, Suginami H, Kuramori R, Yabuta M, Suginami R, Taniguchi F. Four subtypes of adenomyosis assessed by magnetic resonance imaging and their specification. Am J Obstet Gynecol 2012;207:114.e1–7.
Reply: Thank you. We have revised the manuscript as suggested. (Line 44-47: "Four subtypes of adenomyosis assessed by MRI were classified below: subtypes I-III were suggested as a cause of direct endometrial invasion, ectopic endometriotic invasion from the outside, and de novo metaplasia, respectively, whereas subtype IV was a heterogeneous mixture of far advanced disease [8].")

Comment 2: 2.2. Theories and potential mechanisms of adenomyosis The recent unique "Axonemal Alteration Hypothesis" (Khan KN, Fujishita A, Suematsu T, Ogawa K, Koshiba A, Mori T, et al. An axonemal alteration in apical endometria of human adenomyosis. Hum Reprod 2021;36:1574–89) may be introduced.
Reply: Thank you. The axonemal alteration hypothesis provides a novel concept of adenomyosis-associated infertility. Thus, the study was introduced in section 3. Pathologic mechanism of adenomyosis-associated infertility. (Line 334-338: "Recently, Khan et al. [107] demonstrated microvilli malformation and axonemal arrangement disruption in the apical endometria derived from women with focal and diffuse adenomyosis. These ultra-structural endometrial abnormalities in response to tissue inflammation with abundant macrophage infiltration may negatively affect fertility outcomes in women with symptomatic adenomyosis." Line 294: "Axonemal alteration in apical endometria" was also added in Figure 3.)

Comment 3: 2.2. Theories and potential mechanisms of adenomyosis Recent studies suggest that Fusobacterium infection in endometrial stromal fibroblasts is a potential cause of endometriosis, an adenomyosis-related disease (Muraoka A, Suzuki M, Hamaguchi T, Watanabe S, Iijima K, Murofushi Y, Shinjo K, Osuka S, Hariyama Y, Ito M, Ohno K, Kiyono T, Kyo S, Iwase A, Kikkawa F, Kajiyama H, Kondo Y. Fusobacterium infection facilitates the development of endometriosis through the phenotypic transition of endometrial fibroblasts. Sci Transl Med. 2023 Jun 14;15(700):eadd1531
Microbiome studies that compared the gut, uterine cavity, and vaginal cavity between adenomyosis and non-adenomyosis patients can be quoted and discussed.
Reply: Thank you. Microbiome studies were added in section 2.3 Pathogenesis of adenomyosis.
(Line 276-281: "2.3.6 Microbiota composition Alteration in the microbiota composition has been postulated in the pathogenesis of adenomyosis by affecting the host's epigenetic, immunologic, metabolic, and biochemical functions. Distinct bacterial taxa were identified in the endometrial, vaginal, and gut microbiota of adenomyosis patients compared with controls, suggesting a potential link between microbiota profile and adenomyosis [91].")

Comment 4: 2.2. Theories and potential mechanisms of adenomyosis The relationship between adenomyosis and chronic endometritis (Khan KN, Fujishita A, Ogawa K, Koshiba A, Mori T, Itoh K, et al. Occurrence of chronic endometritis in different types of human adenomyosis. Reprod Med Biol 2022;21:e12421. ; Kitaya K, Matsubayashi H, Yamaguchi K, Nishiyama R, Takaya Y, Ishikawa T, et al. Chronic endometritis:Potential cause of infertility and obstetric and neonatal complications. Am J Reprod Immunol 2016;75:13–22.) may be referred.
Reply: Thank you. The studies of chronic endometritis were discussed in section 3. Pathologic mechanism of adenomyosis-associated infertility. (Line 339-342: "In addition, women with adenomyosis have a higher concurrent rate of chronic endometritis [108]. A variable presence of chronic endometritis in women with adenomyosis could be an additional factor that worsens the endometrial receptivity by distorting the functional and structural integrity of the endometrium [109].")

Comment 5: 4.3. Uterus sparing surgery The article on adenomyomectomy by Nishida M et al, (Masato Nishida, Katsumi Takano, Yuko Arai, Hirokazu Ozone, Ryota Ichika Fertil Steril. 2010 Jul;94(2):715-9. Conservative surgical management for diffuse uterine adenomyosis) may be worth introducing here as it discloses the surgical methodology with detailed figures.

Reply: Thank you. We have revised the manuscript as suggested. (Line 511: The study was cited as reference 157.)

Thank you again for all the thoughtful inputs to strengthen this manuscript.

Yours sincerely,

Liang-Hsuan Chen, M.D. & Hsien-Ming Wu, M.D., Ph.D.

Reviewer 2 Report

Comments and Suggestions for Authors

A well written and comprehensive review discussing about the management options for women with adenomyosis. The authors were able to nicely present the different theories on the aetiology of this complex disease and summarize the current/existing approaches to treat these patients. Overall, the review was concise and supported with relevant references.  

Author Response

We thank the reviewer for taking the time and effort to review the manuscript.